# Comparative Evaluation of Diagnostic Tests for Brucellosis in Humans and Animals: A Meta-Analytical Approach

**DOI:** 10.3390/vetsci12070638

**Published:** 2025-07-03

**Authors:** Shahid Nazir, Muhammad Farooq, Raheel Khan, Aman Ullah Khan, Ali Husnain, Muhammad Ali Hassan, Hosny El-Adawy, Heinrich Neubauer

**Affiliations:** 1Department of Pathobiology, University of Veterinary and Animal Sciences, Lahore, Sub-Campus Jhang, Jhang 35200, Pakistan; shahidnazir13100@gmail.com (S.N.); alihassangoraya423@gmail.com (M.A.H.); 2Department of Clinical Sciences, University of Veterinary and Animal Sciences, Lahore, Sub-Campus Jhang, Jhang 35200, Pakistan; muhammad.farooq@uvas.edu.pk (M.F.); khan.raheel363@gmail.com (R.K.); 3Department of Theriogenology, University of Veterinary and Animal Sciences, Lahore 54000, Pakistan; 4Institute of Bacterial Infections and Zoonoses, Friédrich-Loeffler-Institut, 07743 Jena, Germany; heinrich.neubauer@fli.de

**Keywords:** *Brucella*, brucellosis, diagnosis, tests comparison, meta-analysis

## Abstract

Brucellosis is a bacterial zoonosis that affects both humans and animals, causing significant public health concerns and economic losses, particularly in countries where it is endemic. To prevent the spread of disease, early and accurate diagnosis is critical. Multiple diagnostic tests are used worldwide; however, no single test is sufficient in different epidemiological contexts, and for all relevant host species. This study analyzed data from 135 articles published between 2013 and 2023 that contained relevant data of nearly 20,000 humans and 64,000 animals. The objective of this study was to determine which tests demonstrated superior detection rates when applied simultaneously to the same number of samples. The results of this study revealed that primary binding assays had higher comparative detection rates than the Rose Bengal plate test (RBPT), a commonly used screening test, for diagnosing brucellosis in humans. Slow agglutination tests had lower detection rates than the RBPT, both in humans and cattle. Similarly, the complement fixation test (CFT) had a lower comparative detection rate than the RBPT, both in cattle and sheep. This study will help veterinarians, doctors, and public health authorities in selecting the most suitable tests across different species and epidemiological settings for effective diagnosis and control of brucellosis.

## 1. Introduction

Brucellosis is a highly contagious and often neglected zoonotic bacterial disease that significantly impacts livestock production and poses a considerable public health burden globally.

Brucellosis is caused by Gram-negative, facultative intracellular coccobacilli of the genus *Brucella* [1]. This genus contains twelve well-characterized *Brucella* species characterized by specific host preferences. Three species caused significant economic losses and public health hazards, i.e., *B. abortus* (cattle, buffaloes, and camels), *B. melitensis* (goats, sheep, and camels), and *B. suis* (pigs and camels) [2,3]. *Brucella canis* is also recognized as a potential cause of human brucellosis; however, the disease remains poorly characterized due to its non-specific clinical presentation and the limited research into the condition [4].

Although brucellosis has worldwide distribution, it has been successfully controlled in domestic animals in developed countries, such as Australia, New Zealand, Canada, and Western Europe, through eradication programs. However, in underdeveloped regions of Africa, Asia, the Mediterranean basin, South America, and Latin America, brucellosis remains endemic in humans, farm animals, and wildlife [5,6]. In developing countries, an estimated 3.5 billion people are at risk of getting infected with one or more *Brucella* spp. [7].

Brucellae are shed by infected animals mainly via secretions and excretions such as milk, semen, vaginal discharge, placental and birth fluids. These are sources of infection for susceptible animals. Direct or indirect transmission involves ingestion, inhalation, or contact, or may occur during mating [8,9]. Brucellae may penetrate mucosa or submucosa, e.g., the conjunctiva. Vertical transmission involves congenital and prenatal infection, when pathogens are passed from mother to offspring [10]. Wildlife species, such as bats, voles, bison, boars, hares, elk, and foxes, serve as reservoirs for *Brucella* [11,12]. Ticks and lice have also been suggested as potential reservoirs or vectors, although evidence of this remains limited [13,14]. Brucellosis results in a variety of clinical signs in animals with deep economic impacts, e.g., infertility, abortion, retention of fetal membranes, and prolonged calving intervals resulting in reduced productivity of a herd, loss in milk and meat production, reduced weight gain, lost draft power, and culling [15,16,17].

Humans acquire brucellosis primarily through the ingestion of unpasteurized dairy products or by direct contact with infected animals, particularly with excretions, aborted fetuses, etc., during parturition [18]. Occupational infections are commonly seen in veterinarians, farmers, hunters, butchers, and laboratory personnel as they are more often exposed to diseased animals or their products [19]. The most common symptom in the acute stage is undulant fever with chills, headache, fatigue, arthralgia, splenomegaly, hepatomegaly, jaundice, and lymphadenopathy. Other symptoms are anorexia, weakness, diarrhea, asthenia, and malaise [20,21,22]. The disease tends to become chronic with chronic fatigue, depression, uveitis, episcleritis, and spondylitis [23]. Localized infections may be seen in all organs after bacteremia. Infection poses a high risk to pregnant women and unborn fetuses. Evidence suggests that *Brucella* spp. (*B. ceti* and *B. pinnipedialis*) found in marine mammals also possesses zoonotic potential. They can be transmitted to humans through the consumption of raw or undercooked seafood, potentially leading to neurobrucellosis, with signs of meningoencephalitis, myelitis, and cerebral involvement [24,25,26].

The gold standard approach for diagnosing brucellosis is isolation and identification of the *Brucella* organism from cultures of clinical specimens such as blood, tissues, or other body fluids [27]. However, it has certain limitations, such as the availability of samples containing bacteria, increased risk for laboratory personnel during handling, and being time-consuming and labor-intensive [28]. The definitive diagnosis depends upon the detection of *Brucella* or its DNA through molecular assays, such as polymerase chain reaction (PCR) [29]. However, these assays are more expensive, require specialized equipment, and may not necessarily represent an active infection [30]. Indirect diagnosis of brucellosis is usually made through various serological assays that detect antibodies against *Brucella* antigens. These tests include rapid agglutination tests, such as the Rose Bengal plate test (RBPT), slow agglutination tests, such as the standard tube agglutination test (SAT), the complement fixation test (CFT), and primary binding assays, such as enzyme-linked immunosorbent assays (ELISAs) and the fluorescent polarization assay (FPA) [31]. The agglutination tests are compromised by unacceptably high numbers of false-negative and false-positive results, cross-species reactivity, and lack of common criteria for interpretation [31,32]. ELISAs are widely used for the screening and confirmation of brucellosis due to their high sensitivity and specificity, with competitive ELISA (cELISA) offering improved specificity over indirect ELISA (iELISA) by minimizing cross-reactions, especially with *Yersinia* species, though sometimes at the cost of reduced sensitivity [33,34].

Due to the lack of specific clinical signs, laboratory testing is currently the cornerstone of brucellosis diagnosis. Choosing an appropriate diagnostic tool is essential for effective disease surveillance and control. However, due to variations in diagnostic sensitivity and specificity, no single test is suitable for different epidemiological contexts or host species, and the performance of tests may vary according to the stage of disease [35,36]. Although previous systematic reviews and meta-analyses have contributed significantly to the understanding of the diagnostic accuracy of tests used for brucellosis. They primarily focused on individual test characteristics (sensitivity/specificity estimates using variable reference standards), specific test formats, or single host species [37,38]. We hypothesized that certain diagnostic methods (e.g., iELISA or PCR) would demonstrate significantly higher detection rates than others (e.g., RBT or SAT) when applied to the same sample sets (parallel testing). By incorporating data from 135 studies, this study provides a broader comparative synthesis by evaluating multiple diagnostic tests across a wide range of host species. By calculating and comparing the relative risk (RR) of test positivity (the likelihood of a test detecting brucellosis compared to another test), this meta-analysis is intended to identify which tests demonstrate superior detection rates when applied simultaneously to the same number of samples. This aims to inform the selection of optimal screening tools, improve the consistency of diagnostic practices, and ultimately support more effective surveillance and control strategies for brucellosis in both animals and humans.

## 2. Materials and Methods

### 2.1. Review Assessment Protocol

This systematic review was performed according to the established principles outlined in the Cochrane Handbook [39], and transparency was maintained by adhering to the Preferred Reporting Items for Systematic Reviews and Meta-Analyses (PRISMA) guidelines as described by [40]. The review process involved thorough database searches for selecting relevant articles, followed by a rigorous selection based on predefined criteria. The selected articles were then analyzed for their relevance to the research question, and data were extracted, screened, and analyzed following standardized procedures.

### 2.2. Literature Search Strategy

A systematic and comprehensive search strategy was implemented to identify relevant research articles on the global prevalence and geographical distribution of brucellosis published in English between 2013 and 2023. The major platforms searched include PubMed “https://pubmed.ncbi.nlm.nih.gov/ (accessed on 15 April 2023)”, Science Direct “https://www.sciencedirect.com/ (accessed on 20 April 2023)”, Web of Science “https://www.webofscience.com/wos/woscc/advanced-search (accessed on 22 April 2023)”, and Scopus “https://www.scopus.com/search/form.uri?display=basic#basic (accessed on 25 April 2023)”. In addition to electronic databases, a supplementary search was conducted in repositories for theses and dissertations; “http://opac.uvas.edu.pk/ (accessed on 27 April 2023)”, at UVAS Pakistan and “https://www.proquest.com (accessed on 28 April 2023)”, in the USA. All electronic databases were comprehensively searched from 15 April to 28 April 2023, to identify the relevant literature on brucellosis prevalence. The search strategy employed a combination of Medical Subject Headings (MeSHs) terms and keywords related to brucellosis prevalence in humans and animals. The keywords were searched by using Boolean operators (AND, OR) to clarify the search, and ensuring that the search captured relevant studies based on the Title, Abstract, or Keywords. The keywords included were ((“Brucellosis” OR “*Brucella*” OR “Malta fever” OR “Mediterranean fever” OR “Mediterranean remittent fever” OR “Undulant fever” OR “Gibraltar fever” OR “Rock fever” OR “Neapolitan fever”) AND (“Molecular detection” OR “Diagnosis” OR “Prevalence” OR “Seroprevalence” OR “Surveillance” OR “Epidemiological survey” OR “Serology” OR “Serodiagnosis” OR “PCR”) AND (“Human” OR “Animals” OR “Buffalo” OR “Cow” OR “Cattle” OR “Sheep” OR “Goat” OR “Dog” OR “Horse” OR “Camel” OR “Pig” OR “Wildlife” OR “Marine”)) for database search.

### 2.3. Inclusion and Exclusion Criteria

Following the initial electronic database search, a two-tiered screening process was implemented. In the first step, duplication was removed. Initial screening of the studies was performed by their titles; subsequently, abstracts were reviewed to select studies published from 2013 to 2023 that investigated brucellosis prevalence either in humans or in animals. In the second stage, an evaluation of full-text articles was conducted to verify the use of a validated diagnostic method. The eligibility criteria were established based on the hypotheses of the meta-analysis. To be included, studies were required to meet the following criteria: (1) they were required to be peer-reviewed, original research articles in English language; (2) published between 1 January 2013 and 28 April 2023 to ensure the most recent data; (3) employed cross-sectional or observational designs at a minimum; (4) reported the total sample size and the number of animals testing positive for brucellosis using a validated diagnostic method; and (5) articles where samples were tested for brucellosis with at least 2 diagnostic tests in parallel. Articles published in other languages and reviews were excluded. Subsequently, articles that satisfied this inclusion criterion were selected for meta-analysis.

### 2.4. Research Articles Included

Figure 1 illustrates a PRISMA flow diagram outlining the data collection process for the meta-analysis. After the initial search, a total of 6536 records were identified that were published between 2013 and 2023, comprising 6520 records from databases and 16 records from repositories for theses and dissertations. From these records, 1018 studies were removed because of duplication, and 5383 studies were excluded based on predefined eligibility criteria. Ultimately, a total of 135 records, involving 328 comparisons that contained relevant data of 19,921 humans and 64,145 animal samples for brucellosis prevalence, were included. A comprehensive list of records included in the meta-analysis is provided in Appendix A.

### 2.5. Data Extraction and Quality Assessment

Two authors independently extracted data from all the eligible studies. All relevant information was recorded in a Microsoft Excel spreadsheet, including the first author’s name, year of publication, study design, country of study, geographical location, animal species, animal species category, sampling information, sample size, diagnostic methods used, the number of samples that were found to be positive for *Brucella* in each test, and the *Brucella* species under investigation. The extracted data were then thoroughly reviewed by two other reviewers to ensure the accuracy of the collected information before further analysis. The methodological quality of all included studies was assessed using the Joanna Briggs Institute (JBI) critical appraisal tool [41]. The checklist consists of eight questions evaluating the clarity of inclusion criteria, measurement reliability, validity of outcome measures, confounding factors, and appropriate statistical analysis. Each study was independently assessed by two reviewers and scored using “Yes” or “No” responses for each criterion. Studies scoring positively on ≥6 out of 8 questions were classified as high-quality, ≥4 and 5 as moderate-quality, while those scoring below 4 were designated as low-quality. The quality score and the corresponding quality level for each study are provided in Appendix A.

### 2.6. Statistical Analysis

The classical meta-analysis was conducted by using the metabin() function in the “meta” package of the R statistical software (version 4.4.1) to calculate pooled relative risk (RR) estimates. RR, a commonly used statistical measure, was selected as the summary measure due to its easy and direct interpretation and suitability for within-study comparisons. It enables a direct comparison of the likelihood of a diagnostic test detecting disease relative to a reference test, and provides a quantitative estimate of how much more or less likely a test is to yield a positive result compared to another.

#### 2.6.1. Reference Standard

Although historically regarded as the gold standard for its high specificity, culture isolation is now considered a suboptimal reference method due to its relatively low sensitivity [37]. Furthermore, a majority of studies included in the meta-analysis did not use culture as a diagnostic test. In this meta-analysis, the Rose Bengal plate test (RBPT) was used as a control or reference for comparisons, whereas the other test was considered experimental. The RBPT is generally considered a rapid screening test, and 70% of the studies in this dataset reported RBPT as one of the tests applied. Where RBPT was not performed, the following tests in a sequence were considered as a control test: rapid agglutination tests [Card test, Brucella-buffered acidified plate antigen test (BAPA), latex agglutination test (LAT)], slow agglutination tests [milk ring test (MRT), standard agglutination test (SAT), serum plate agglutination test (SPAT), Brucellacapt], primary binding assays (IELISA, c-ELISA), precipitation tests [agar gel immunodiffusion test (AGID)], staining techniques [immunohistochemistry (IHC)], and culture. The categorization of the different diagnostic tests according to [33,42], with little modification, is shown in Appendix A. The subgroup analyses were conducted by grouping the comparisons based on the diagnostic tests’ classification, as shown in Appendix A. When the control was RBPT, the comparisons were grouped based on experimental tests. In contrast, where RBPT was not reported, the comparisons were grouped based on the control test because of multiple control tests. In both cases, the RR (the likelihood of a test detecting brucellosis compared to another test) was estimated for the experimental tests.

#### 2.6.2. Heterogeneity Assessment

Due to the high heterogeneity among the included studies, a random-effects model with the inverse variance method was applied for all analyses. This approach accounts for between-study variability and typically produces wider confidence intervals, providing a more conservative estimate of the central performance measures. Subgroup differences were tested using the Chi-square test for subgroup differences (χ^2^ test), and heterogeneity was quantified using I^2^ statistics and τ^2^. The measure I^2^ is the transformation of the square root of χ^2^ heterogeneity statistics divided by the degree of freedom and is a measure of variation in RR beyond chance among comparisons included in the meta-analysis.

#### 2.6.3. Graphical Summaries

The forest plots were generated for each species separately to estimate the RR of different diagnostic methods. The funnel plots were generated to assess potential publication and reporting bias among studies included in this meta-analysis.

In the forest plot, for each comparison, the point estimate (RR) and respective 95% confidence intervals (CIs) are represented by a black point and its associated horizontal lines, respectively. The central vertical black line represents an RR that is equal to 1, and the points located to the left of the central vertical black line indicate the decrease, whereas the points located to the right represent the increase in the relative risk compared to the control. The term “event” in the forest plot refers to the number of samples testing positive out of the total number of samples tested within a group. Each gray box around the point estimate represents the weight of the comparison; the larger the box, the greater the comparison contribution to the overall estimate. The weight that each comparison contributed is shown in the right-hand column. A diamond represents the overall effect at the bottom of the plot, aggregating the RR of all included studies adjusted by the random effect models.

A minimum criterion for test comparison was the availability of at least three valid comparisons per species or group. The number of comparisons for each species is mentioned below the respective forest plots in the Results section.

#### 2.6.4. Subgroup Meta-Analysis by Study Quality

To evaluate the impact of methodological quality on diagnostic test comparisons, subgroup meta-analyses were performed to stratify studies by their quality level (high vs. low/moderate). This was conducted separately for humans, cattle, goats, and sheep, using only studies that employed RBPT as the reference.

#### 2.6.5. Sensitivity Analysis

To assess the robustness of results, a sensitivity analysis was performed by restricting the meta-analysis to high-quality studies only within each species group. Subgroup meta-analyses were again conducted within these high-quality datasets based on the experimental test group variable, allowing comparison of different diagnostic tests against RBPT. Forest plots were generated to visually compare the magnitude and direction of the pooled estimates and to examine the consistency of findings across subgroups.

Statistical significance was considered for *p* ≤ 0.05 and a tendency at 0.05 < *p* ≤ 0.10.

### 2.7. Publication Bias

Funnel plots (species-wise) depicting the publication bias among all studies included in the meta-analysis are presented in Appendix A. In the funnel plot, the effect size of each study is plotted on the horizontal axis and the standard error (SE) on the vertical axis.

## 3. Results

### 3.1. Comparison of Diagnostic Tests for Brucellosis

A summary of RR estimates for diagnostic test groups compared to RBPT across various species is presented in Table 1. The heatmap depicts the RR of various diagnostic tests compared to RBPT for diagnosing brucellosis across multiple host species (Figure 2).

#### 3.1.1. Human Medicine

Comparison with RBPT

The forest plot represents the relative risk (RR) of six different test groups compared with RBPT for diagnosing brucellosis in humans (Figure 3). Overall, the RR for diagnosing brucellosis when compared to RBPT did not differ significantly [RR (95% CI) = 1.19 (0.88–1.61), *I*^2^ = 91%]. In subgroup analysis, rapid agglutination tests had a significantly higher RR compared to RBPT [RR (95% CI) = 3.43 (1.78–6.59), *I*^2^ = 18%]. Similarly, primary binding assays also had a significantly higher RR [RR (95% CI) = 1.75 (1.35–2.26), *I*^2^ = 73%]. In contrast, slow agglutination tests demonstrated a significantly lower RR [RR (95% CI) = 0.68 (0.48–0.96), *I*^2^ = 90%]. The culture, the complement fixation test (CFT), and PCR did not differ significantly from RBPT.

Comparison across test groups

The forest plot represents the RR of various tests compared to three different test groups for diagnosing brucellosis in humans (Figure 4). The subgroup analyses revealed that experimental tests (Coombs, 2-ME, ELISA, culture, and PCR) had significantly lower RR compared to slow agglutination tests [RR (95% CI) = 0.68 (0.50–0.94), *I*^2^ = 82%]. However, the RRs did not differ significantly when experimental tests were compared with rapid agglutination tests and primary binding assays.

#### 3.1.2. Veterinary Medicine

##### Cattle

Comparison with RBPT

The forest plot depicts the RR of five different test groups compared with RBPT for diagnosing brucellosis in cattle (Figure 5). Overall, the RR for diagnosing brucellosis when compared to RBPT did not differ significantly [RR (95% CI) = 0.97 (0.81–1.17), *I*^2^ = 86%]. In subgroup analysis, slow agglutination tests had a significantly lower RR compared to RBPT [RR (95% CI) = 0.41 (0.25–0.68), *I*^2^ = 96%]. The CFT also demonstrated a slightly but significantly lower RR [RR (95% CI) = 0.97 (0.94–0.99), *I*^2^ = 9%]. However, the primary binding assays, PCR, and rapid agglutination tests did not differ significantly from RBPT.

Comparison across test groups

The forest plot represents the RR of various tests compared with five different test groups (Appendix A). The subgroup analyses revealed that the RR did not differ significantly when experimental tests were compared with the control. The control groups included rapid agglutination tests, slow agglutination tests, primary binding assays, staining, and culture.

##### Buffaloes

Comparison with RBPT

The forest plot represents the RR of five different test groups compared with RBPT for diagnosing brucellosis in buffaloes (Appendix A). Overall, the RR did not differ significantly between the experimental tests and RBPT [RR (95% CI) = 1.11 (0.94–1.32), *I*^2^ = 26%]. The subgroup analyses revealed that the primary binding assays, PCR, rapid agglutination tests, slow agglutination tests, and CFT did not differ significantly from RBPT.

Comparison across test groups

The forest plot represents the RR of various tests compared to three different test groups (Appendix A). The subgroup analyses revealed that the RR did not differ significantly when experimental tests were compared with the control. The control groups included rapid agglutination tests, slow agglutination tests, and primary binding assays.

##### Goats

Comparison with RBPT

The forest plot represents the RR of five different test groups compared with RBPT (Figure 6). Overall, the RR did not differ significantly compared to RBPT [RR (95% CI) = 1.02 (0.83–1.26), *I*^2^ = 58%]. The subgroup analyses revealed that the primary binding assays, CFT, slow agglutination tests, precipitation tests, and PCR did not differ significantly from RBPT.

Comparison across test groups

The forest plot represents the RR of various tests compared to four different test groups (Appendix A). The subgroup analyses revealed that the RR did not differ significantly when experimental tests were compared with the control. The control groups consisted of slow agglutination tests, primary binding assays, staining, and culture.

##### Sheep

Comparison with RBPT

The forest plot represents the RR of five different test groups compared with RBPT (Figure 7). Overall, the RR did not differ significantly compared to RBPT [RR (95% CI) = 0.97 (0.86–1.09), *I*^2^ = 76%]. In subgroup analyses, PCR had a significantly lower RR compared to RBPT [RR (95% CI) = 0.56 (0.32–0.98), *I*^2^ = 42%]. The CFT also demonstrated a slightly but significantly lower RR [RR (95% CI) = 0.97 (0.95–0.99), *I*^2^ = 0%]. However, primary binding assays, slow agglutination tests, and precipitation tests did not differ significantly from RBPT.

Comparison across test groups

The forest plot represents the RR of various tests compared with five different tests (Appendix A). The subgroup analyses revealed that the RR did not differ significantly when experimental tests were compared with the control. The control groups included rapid agglutination tests, slow agglutination tests, primary binding assays, staining, and culture.

##### Camels

Comparison with RBPT

The forest plot represents the RR of three different test groups compared with RBPT (Figure 8). Overall, the RR did not differ significantly compared to RBPT [RR (95% CI) = 0.96 (0.70–1.32), *I*^2^ = 73%]. The subgroup analyses revealed that the CFT had a significantly higher RR compared to RBPT [RR (95% CI) = 1.99 (1.51–2.62), *I*^2^ = 0%]. However, the primary binding assays and slow agglutination tests did not differ significantly from RBPT.

##### Multi-Species Analysis (Ruminants)

Comparison with RBPT

The forest plot represents the RR of four different test groups compared with RBPT (Appendix A). Overall, the RR did not differ significantly compared to RBPT [RR (95% CI) = 0.94 (0.76–1.16), *I*^2^ = 88%]. The subgroup analyses revealed that the CFT, primary binding assays, slow agglutination tests, and PCR did not differ significantly compared to RBPT.

Comparison across test groups

The forest plot represents the RR of various tests compared with three different tests (Appendix A). The subgroup analyses revealed that the RR did not differ significantly when experimental tests were compared with the control. The control groups were agglutination slow tests, primary binding assays, and culture.

##### Pigs

Comparison with RBPT

The forest plot represents the RR of two different test groups compared with RBPT (Appendix A). Overall, the RR did not differ significantly compared to RBPT [RR (95% CI) = 1.22 (0.60–2.46), *I*^2^ = 90%]. The subgroup analyses revealed that the primary binding assays and CFT did not differ significantly from RBPT.

##### Dogs

Comparison with RBPT

The forest plot represents the RR of five different test groups compared with RBPT (Appendix A). Overall, the RR for diagnosing brucellosis was significantly higher compared to RBPT [RR (95% CI) = 1.70 (1.22–2.35), *I*^2^ = 48%]. However, the subgroup analyses revealed that the precipitation tests, CFT, PCR, slow agglutination tests, and primary binding assays did not differ significantly from RBPT.

Comparison across test groups

The forest plot represents the RR of various diagnostic tests compared with five different diagnostic test groups (Appendix A). The subgroup analyses revealed that the RR did not differ significantly when experimental tests were compared with the control. The control groups consisted of rapid agglutination tests, primary binding assays, precipitation tests, staining, and culture.

The analysis was not performed for wildlife species (bison, fox, rodents), marine species (dolphin, walrus), or horses because they had fewer than three test comparisons per category.

### 3.2. Study Quality Assessment

The bar chart illustrates the distribution of study quality based on the JBI critical appraisal checklist (Figure 9a). Out of 135 included studies, 97 (72%) were classified as high-quality, while the remaining 38 (28%) were rated as moderate-quality. The assessment did not identify any studies as being of low quality. The stacked bar plot summarizing responses to the eight JBI checklist questions revealed that the highest proportion of “Yes” responses was observed for Q3: “Was the exposure measured in a valid and reliable way?”. In contrast, the lowest proportion of positive responses was recorded for Q6: “Were strategies to deal with confounding factors stated?” (Figure 9b).

### 3.3. Subgroup Meta-Analysis by Study Quality

The forest plots represent the subgroup meta-analysis by study quality in humans, cattle, goats, and sheep (Appendix A). Subgroup analyses stratified by study quality showed minimal differences in pooled estimates across species. In humans, goats, and sheep, the study quality did not significantly influence the pooled estimates, as the test for subgroup differences was statistically non-significant (*p* > 0.05). However, in cattle, the difference between high- and moderate-quality studies was statistically significant (*p* = 0.02), indicating that study quality contributed to the observed heterogeneity.

### 3.4. Sensitivity Analysis

The forest plots depict the sensitivity analyses based on high-quality studies in humans, cattle, goats, and sheep (Appendix A). The pooled RR estimates remained consistent with the main analyses across all species. The overall RR was slightly elevated in humans, slightly lower in cattle, and remained similar in both sheep and goats.

## 4. Discussion

Brucellosis is a transboundary animal disease classified as a category B infectious disease by the World Organization for Animal Health (WOAH), emphasizing its global importance as a serious health threat to animals and humans alike [137]. Accurate diagnosis of brucellosis is crucial for its effective control, eradication, and surveillance and requires an integrated approach. Multiple diagnostic strategies have been recommended depending upon specific epidemiological contexts. Serial testing (usually a screening test of high sensitivity followed by a confirmatory test of high specificity) is typically employed in low-prevalence areas, pre-movement testing of uninfected herds, and eradication programs. This strategy enhances specificity and minimizes false positives, thereby preventing unnecessary culling or restrictions. In contrast, parallel testing is generally recommended in high-prevalence areas or during outbreak investigations, as it maximizes the sensitivity and reduces the likelihood of false negatives [138,139,140]. This meta-analysis aims to determine the relative risk (RR) of different diagnostic tests used for diagnosing brucellosis in humans and animals, and highlights which tests perform better in parallel testing contexts for improved disease detection. We have further highlighted the strengths and limitations of each diagnostic approach to guide evidence-based decision-making in diverse field settings.

According to the findings of this meta-analysis, primary binding assays detected a higher proportion of positive samples (a higher comparative detection rate) compared to RBPT when both were applied to the same sample sets (parallel testing) in humans. A similar trend was observed in cattle, buffaloes, goats, sheep, and pigs, in which these assays outperformed RBPT in terms of detection. This difference in sensitivity between the primary binding assays and the RBPT for diagnosing brucellosis may be attributed to several factors. Generally, the primary binding assays have higher sensitivities and specificities compared to RBPT [33,141]. As described by [142], the ELISA can be efficiently used to diagnose brucellosis in humans and has a higher sensitivity than RBPT. In the early stages of infection, IgM antibodies predominate, and their strong agglutination capability makes the RBPT, which relies on agglutination, particularly effective for diagnosing the infection during this phase. As the infection progresses to the chronic stage, IgG and non-agglutinating antibodies become more prevalent than agglutinating ones, which can result in the disease going undetected by agglutination tests such as RBPT [143]. All *Brucella* species, except *B. canis* and *B. ovis*, possess a smooth lipopolysaccharide (S-LPS) in their outer cell walls, characterized by an immunodominant O-polysaccharide (OPS) component [144]. *Brucella canis* and *B. ovis* lack OPS and their outer surface contains only rough lipopolysaccharide (R-LPS) [145,146]. The RBPT uses S-LPS antigens and is only suitable for detecting antibodies against smooth *Brucella* species [147]. However, the ELISA can detect antibodies against both the smooth and rough *Brucella* species when it is properly designed [140]. Furthermore, the RBPT has limitations in detecting non-agglutinating antibodies such as IgA and certain forms of IgG [148,149]. The ELISA can detect both IgG and IgM if designed appropriately, and the use of an IgG conjugate in ELISAs enables the detection of non-agglutinating antibodies as well. The ELISA gives a detailed profile of *Brucella*-specific antibodies (IgM, IgG, and IgA) and tends to have a higher sensitivity compared to RBPT, making it highly effective for diagnosing both acute and chronic stages of brucellosis and is well-suited for large-scale screening [87,150]. Moreover, the ELISA, as a quantitative assay, can detect lower antibody titers compared to RBPT, and the use of an ELISA reader also provides more objective and consistent results when compared to the visual interpretation of the RBPT. The superior performance of ELISAs is due to the use of S-LPS fragments, thereby reducing the chances of cross-reactions with other Gram-negative bacteria such as *Yersinia enterocolitica* O:9, *Salmonella enterica*, *Escherichia coli* O:116, and O:157, or *Vibrio cholerae* [33]. Some of the primary binding assays may also distinguish between antibodies produced by vaccination and those resulting from infection [146]. The primary binding assays may help to mitigate the prozone effect, which is often observed with acidified antigens in the RBPT [151,152]. This is a phenomenon where an excess of antibodies or antigens can lead to a false-negative result in an immunoassay. This effect can be avoided by the serial dilution of serum. A 1:8 dilution of serum in RBPT, rather than the standard 1:2 dilution, has been shown to enhance the test’s specificity without significantly impacting sensitivity, increasing complexity, time required, or costs [153]. Additionally, in the RBPT, the acidic nature of the antigen may help to suppress the effects of non-specific agglutinins, potentially leading to reduced detection rates. Despite these prevailing limitations, RBPT is still internationally recommended as a screening test for brucellosis in resource-limited countries because it is rapid, cost-effective, easy to perform, and can be conducted in farm settings. Moreover, rapid laboratory diagnostic procedures for the identification of causative agents are important for the timely implementation of appropriate public health countermeasures. While primary binding assays generally offer better performance compared to rapid diagnostic tests, their higher cost and sophisticated equipment needs often limit their use to a second-line option in low-resource settings [154]. The results of these assays should be interpreted in conjunction with culture and/or molecular identification of *Brucella*. As in humans, the occurrence of a rheumatoid factor is responsible for false-positive results in the primary binding assays, particularly ELISAs [27].

This meta-analysis revealed that RBPT has a higher comparative detection rate than slow agglutination tests in cattle and humans. Similarly, in goats, sheep, and camels, RBPT is more effective at detecting brucellosis than slow agglutination tests. This difference in detection between RBPT and slow agglutination tests for diagnosing brucellosis could be attributed to several factors. When the analysis was restricted to high-quality studies in humans, the association was attenuated and no longer statistically significant. This shift suggests that study quality may have influenced the originally observed performance difference in slow agglutination tests in humans. The sensitivity of slow agglutination tests is low compared to RBPT [33,155]. The cut-off titers for the standard tube agglutination test (SAT) in determining positive or negative results depend on factors such as the type of specimen, the antigen used, and whether the region is endemic or non-endemic [150]. For instance, in China, the official diagnostic criterion for brucellosis, as stated by [156], is a titer greater than 1:100 with clear agglutination (>50%), which is lower than the WHO’s recommended cut-off of >1:160. In endemic regions, a cut-off titer of 1/320 is recommended for the SAT [157], while the RBT is sensitive to titers below 1:320 [50]. The SAT is more prone to false-negative results than RBPT due to the nature of the antigen used, the presence of prozone phenomenon, blocking, and non-agglutinating antibodies [153]. However, incorporation of certain modifications, such as the addition of EDTA, 2-mercaptoethanol, or antihuman globulin, can enhance accuracy [158]. Slow agglutination tests are less effective for detecting IgG antibodies [147]. In modified slow agglutination tests, reducing agents are added to break the disulfide bridges of IgM to prevent non-specific reactions. However, this process can also affect some IgG molecules, potentially leading to false-negative results as these antibodies lose their ability to agglutinate [42]. The use of mercapto-ethanol in 2ME tests neutralizes IgM, thereby inhibiting serological responses to *Yersinia enterocolitica* O:9 and other sources of false-positive reactions [159]. Due to their IgM-reducing effect, these tests are not useful during the acute stage of infection. Slow agglutination tests generally exhibit higher specificity compared to RBPT, resulting in fewer false-positive reactions [147,160]. RBPT may sometimes produce false reactions due to fibrin clot formation if samples are incubated for extended periods [42]. The SAT is generally not recommended by the WOAH for diagnosing brucellosis as it is considered inferior to other standard tests. While slow agglutination tests offer better specificity compared to RBPT, they are time-consuming, require labor-intensive protocols, and are prone to false-negative reactions, making them unsuitable for use as screening tests. The findings of this study suggest that the RBPT screening test is preferred to the slow agglutination tests. However, its results must be validated with one of the confirmatory tests, such as primary binding assays, to overcome its limitations and improve diagnostic accuracy.

The results of this meta-analysis illustrate that slow agglutination tests, particularly SAT, have a higher comparative detection rate than various experimental tests, including Coombs, 2-ME, primary binding assays, culture, and PCR in humans. A similar trend has been observed in cattle, goats, and sheep. The SAT is a serological test based on the detection of specific antibodies in a serum sample. While it provides presumptive evidence for infection, it cannot differentiate between active and past infection. The main disadvantages of the SAT are low specificity compared to modified slow agglutination tests, culture, and PCR, and the challenges in interpreting the results, particularly in patients with recurrent exposure to *Brucella* [148]. Additionally, the diagnostic performance of the SAT can be limited by the presence of non-agglutinating antibodies in chronic cases [161]. Unlike the SAT, culture and PCR are direct diagnostic methods that require the presence of *Brucella* bacteria or their genetic material in a sample, respectively.

This meta-analysis revealed that compared to molecular assays (PCR), bacterial culture demonstrated a lower comparative detection rate. This can be ascribed to the fastidious nature of *Brucella* [149]. Isolation failures may arise from a low quantity of viable bacteria in a clinical sample or when the sample is heavily contaminated [162]. Blood cultures are useful only in patients with bacteremia, which may not always occur. Several inhibitor substances, such as anticoagulants, hemoglobin, and host DNA, are present in whole blood [163]. The stage of infection may affect the quantity and localization of the organisms within the different body tissues [164]. During the acute phase of infection, bacterial circulation in the bloodstream often leads to positive blood cultures. However, as the infection progresses to the chronic stage, bacteria may become localized in various body tissues, resulting in negative blood cultures. During the sub-acute stage of disease, culture results are often negative [156]. Additional drawbacks of culture and isolation include that they are time-consuming, labor-intensive, require a BSL-3 laboratory, and pose a significant risk of infection to laboratory workers [165]. To address these challenges, molecular biological techniques, e.g., PCR, are frequently utilized to identify species and characterize strains of *Brucella* [30]. Although molecular tests are generally expected to have high sensitivity for most diseases, they do possess certain drawbacks in brucellosis testing. They may not identify an active infection, as they can detect a low bacterial count caused by DNA from dead bacteria [37]. The sensitivity of conventional PCR can fluctuate based on the sample type; for instance, the peripheral blood sample may not always contain an adequate amount of bacterial DNA for detection, particularly in chronic or latent cases, leading to reduced sensitivity. These limitations can be addressed by the use of real-time PCR, which provides improved analytical sensitivity and can detect lower levels of bacterial DNA more effectively in clinical samples [166,167].

The findings of this meta-analysis indicate that RBPT has a higher detection rate than CFT for diagnosing brucellosis in cattle and sheep. A similar trend is observed in goats, and when samples of different ruminant species are investigated. CFT has lower sensitivity but higher specificity compared to RBPT. As reported by [168], CFT has demonstrated the highest specificity and shown the fewest false positives. The high specificity of CFT is largely due to its detection of IgG antibodies only, as IgM is partially destroyed during the inactivation process [147]. Since CFT primarily detects IgG1 antibodies, the chances of cross-reactivity with other Gram-negative bacteria are minimized. In contrast, CFT exhibited a higher detection rate than RBPT for diagnosing brucellosis in camels. However, the dataset for this particular subgroup meta-analysis was not robust and comprised only two studies. It is important to note that in one of these studies, 60 out of 118 CFT-positive samples tested negative by other assays, including ELISA and PCR, and showed lower titers of 1:10 or 1:20, and rarely 1:40 [3]. CFT is technically a complex test, requiring careful titration of various test reagents and the inclusion of all control reagents in each test set. The test also has other limitations, including the need for heat-inactivated serum samples to prevent anti-complementary activity, a potential for prozone effects, and the subjective interpretation of results. Additionally, challenges such as the direct activation of the complement by serum (anti-complementary activity), inability to utilize hemolyzed serum samples, the need for well-equipped laboratories, and skilled personnel further complicate its application [144].

Generally, the sensitivity of CFT is lower than that of primary binding assays (ELISAs), which may be due to the fact that ELISA utilizes a protein G conjugate that detects immunoglobulin isotypes IgG1 and IgG2, whereas CFT detects IgG1 only. Additionally, ELISA has been found to detect IgG1 in lower amounts compared to CFT [169]. The CFT can yield false-negative reactions, probably due to hindrance caused by IgG2-type antibodies in complement fixation [170]. The ELISA has been reported to be an effective screening test, whether employed independently or in combination with the RBPT [143]. The competitive ELISA (cELISA) has been demonstrated to eliminate some but not all false-positive serological reactions (FPSRs) produced by cross-reacting microorganisms [171]. These assays avoid the effects of non-agglutinable antibodies. The FPA is a simpler and more cost-effective test than CFT and ELISAs, and can even be conducted outside a laboratory setting with relative ease [35,172]. The primary binding assays, such as ELISAs and the fluorescence polarization assay (FPA), offer comparable or superior diagnostic performance to CFT and are technically simpler and more robust, making them suitable options for use as either a screening or confirmatory tool [168].

In the current study, the febrile antigen *Brucella* agglutination test (FBAT) has a higher comparative detection rate than RBPT for diagnosing brucellosis in humans. The higher detection rate of FBAT when compared to RBPT in humans could be due to the following findings: FBAT has poor diagnostic specificity and can lead to false-positive reactions [65,173]. The high incidence of potentially false-positive test results has several implications. FBAT’s low positive predictive value has been identified as a factor contributing to frequent over-diagnosis [174]. The sensitivity and the specificity of FBAT are lower than those of RBPT. In comparison to FBAT, the RBPT offers high diagnostic accuracy and lower costs per sample, making it the preferred test for diagnosing brucellosis in humans [44,65].

The sensitivity analysis and subgroup meta-analyses by study quality indicated that study quality had limited influence on the overall estimates and did not substantially alter the direction or significance of the findings. This consistency across quality strata enhances the robustness and generalizability of the pooled estimates. However, in cattle, study quality significantly influenced the results, implying that methodological rigor may impact the diagnostic performance estimates. Furthermore, the persistence of significant subgroup differences across diagnostic tests indicates that variability in test performance is inherent and not solely attributable to study quality. This highlights the need for careful selection and validation of diagnostic tools in future brucellosis research and surveillance efforts.

## 5. Conclusions

According to the findings of this meta-analysis, primary binding assays should be preferred for large-scale screening of brucellosis in both humans and animals due to their ability to detect more positive cases (a higher comparative detection rate), followed by the RBPT and slow agglutination tests. However, their applicability may vary by host species, epidemiological context, and resource availability. Culture and PCR methods, while definitive, show variable detection rates and were less frequently reported in the included studies. Since no single test proved universally optimal, there is a need for context-specific diagnostic strategies and combined testing approaches. Future studies should aim for standardized reporting of test combinations and relevant contextual factors to enable more tailored and effective diagnostic strategies for brucellosis control.

## 6. Limitations

This meta-analysis, while comprehensive, is subject to certain limitations that should be acknowledged. Firstly, the use of the RBPT as the primary control in most comparisons, rather than bacterial culture, the gold standard for diagnosing brucellosis, is likely to influence the accuracy of relative risk estimations. While RBPT is widely employed due to its simplicity and cost-effectiveness, its limitations in sensitivity and specificity as compared to culture may affect comparative evaluations. Furthermore, there was significant heterogeneity across the majority of our results. We performed subgroup analyses to better understand the sources of heterogeneity; however, due to the limited number of studies in the majority of subgroups, results should be interpreted with caution, particularly those that had fewer than two studies or with small sample sizes. Variability in diagnostic test protocols, interpretation criteria, and reagent sources further contributes to inter-study differences. These factors collectively emphasize the need for cautious interpretation of the findings and underscore the importance of standardized diagnostic practices in future studies.

## Figures and Tables

**Figure 1 vetsci-12-00638-f001:**
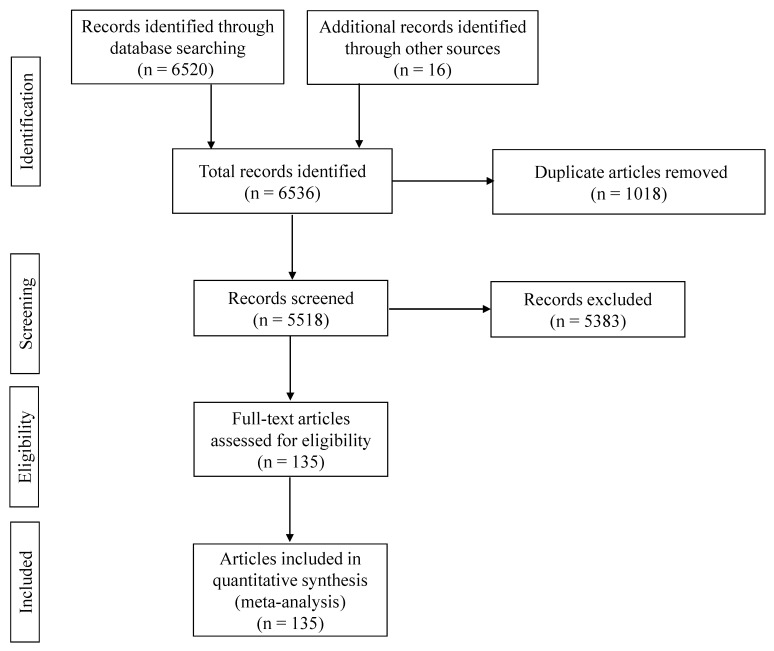
The PRISMA flow diagram [40] of the systematic review from initial search and screening before final selection of publications to be included in the meta-analysis. After initial identification of the records that appeared in each search engine with certain keywords, the titles of the articles were compared across the search engines to remove duplicates. Records were subsequently screened, and articles related to the hypothesis were assessed based on the eligibility criteria; ultimately, one hundred and thirty-five articles were included in the meta-analysis.

**Figure 2 vetsci-12-00638-f002:**
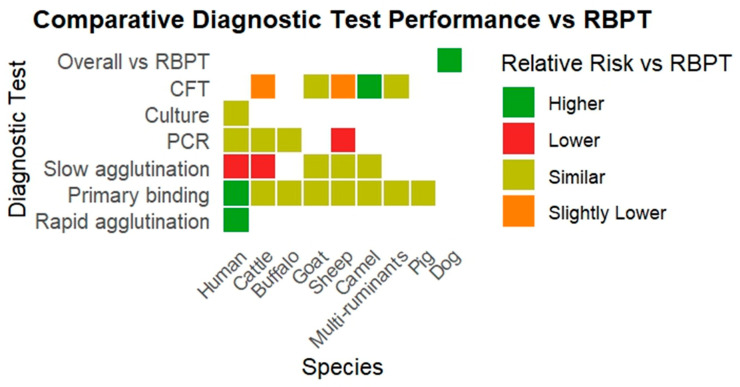
The heatmap depicts the RR of various diagnostic tests compared to RBPT for diagnosing brucellosis across multiple host species.

**Figure 3 vetsci-12-00638-f003:**
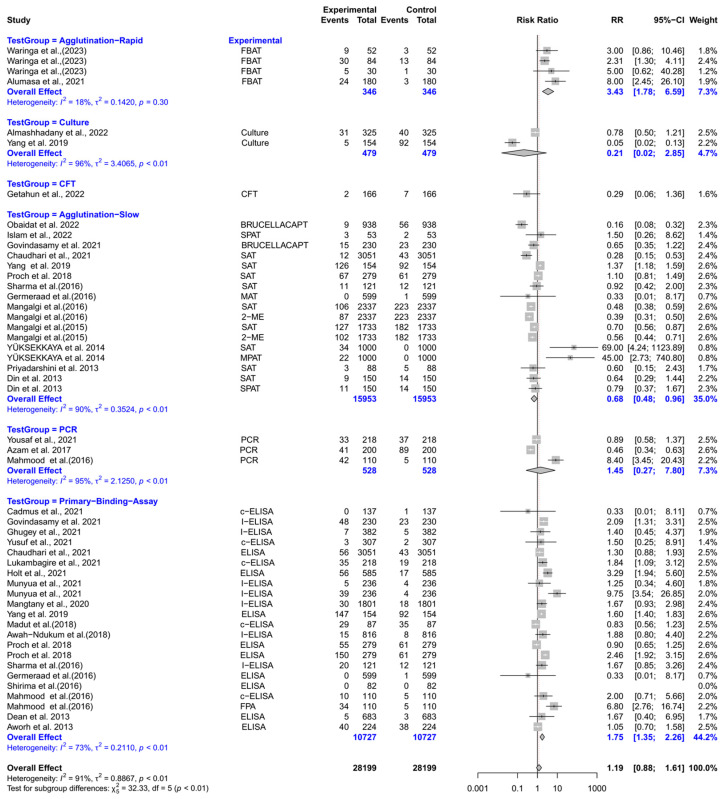
The forest plot represents the relative risk (RR) of six different test groups [experimental] compared with the Rose Bengal plate test (RBPT) [control] for diagnosing brucellosis in humans. A total of 49 comparisons are depicted in the forest plot. One comparison was excluded because it had zero positive events in both groups, i.e., experiment and control. In the forest plot, squares represent the point estimate (RR), and their associated horizontal lines indicate the respective 95% CIs. The size of each square reflects the weight of that comparison in the meta-analysis. The diamond represents the overall effect for all studies or within subgroups. The central vertical black line represents an RR of 1, and the dotted red line indicates the pooled RR from the meta-analysis. Overall, the RR for diagnosing brucellosis when compared to RBPT did not differ significantly [RR (95% CI) = 1.19 (0.88–1.61), *I*^2^ = 91%]. In subgroup analysis, rapid agglutination tests had a significantly higher RR compared to RBPT [RR (95% CI) = 3.43 (1.78–6.59), *I*^2^ = 18%]. Similarly, primary binding assays also had a significantly higher RR [RR (95% CI) = 1.75 (1.35–2.26), *I*^2^ = 73%]. In contrast, slow agglutination tests demonstrated a significantly lower RR [RR (95% CI) = 0.68 (0.48–0.96), *I*^2^ = 90%]. The culture, the complement fixation test (CFT), and PCR did not differ significantly from RBPT [43,44,45,46,47,48,49,50,51,52,53,54,55,56,57,58,59,60,61,62,63,64,65,66,67,68,69,70,71,72,73].

**Figure 4 vetsci-12-00638-f004:**
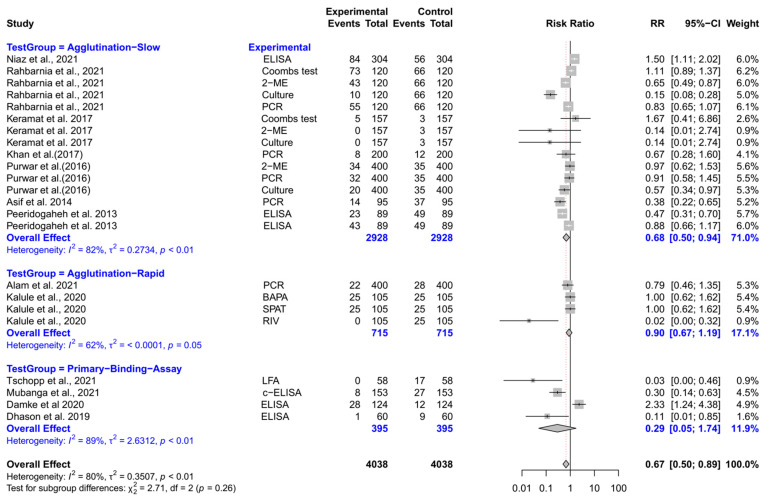
The forest plot represents the relative risk (RR) of various tests [experimental] compared to three different test groups [control] for diagnosing brucellosis in humans. A total of 23 comparisons are depicted in the forest plot. In the forest plot, squares represent the point estimate (RR), and their associated horizontal lines indicate the respective 95% CIs. The size of each square reflects the weight of that comparison in the meta-analysis. The diamond represents the overall effect for all studies or within subgroups. The central vertical black line represents an RR of 1, and the dotted red line indicates the pooled RR from the meta-analysis. The subgroup analyses revealed that experimental tests (Coombs, 2-ME, ELISA, culture, and PCR) had significantly lower RR compared to slow agglutination tests [RR (95% CI) = 0.68 (0.50–0.94), *I*^2^ = 82%]. However, the RRs did not differ significantly when experimental tests were compared with rapid agglutination tests and primary binding assays [74,75,76,77,78,79,80,81,82,83,84,85,86].

**Figure 5 vetsci-12-00638-f005:**
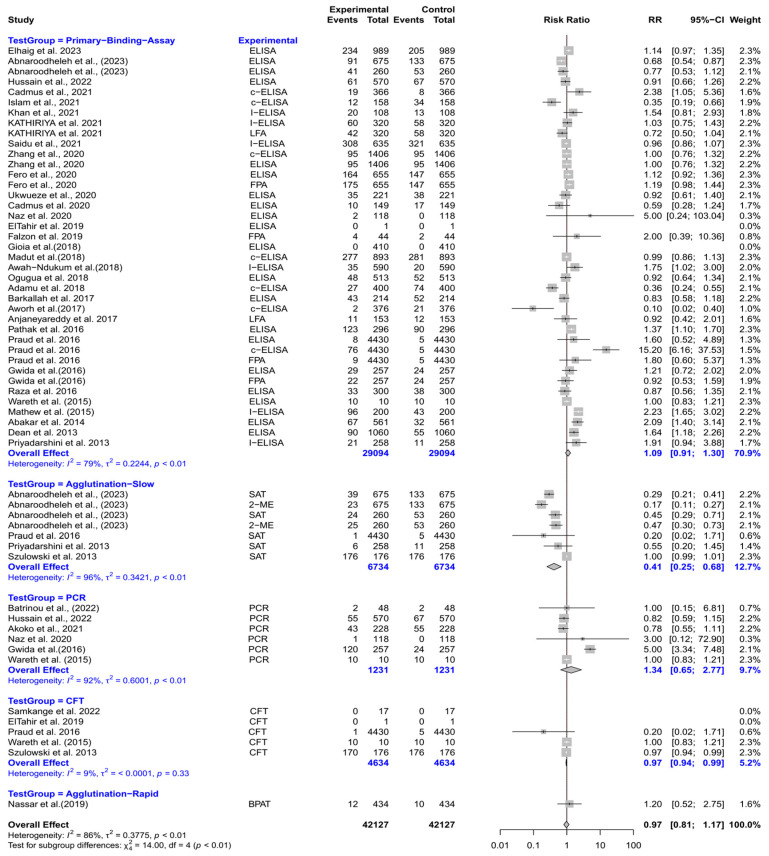
The forest plot depicts the relative risk (RR) of five different test groups [experimental] compared with the Rose Bengal plate test (RBPT) [control] for diagnosing brucellosis in cattle. A total of 58 comparisons are depicted in the forest plot. Four comparisons were excluded because they had zero positive events in both groups, i.e., the experimental group and the control group. In the forest plot, squares represent the point estimate (RR), and their associated horizontal lines indicate the respective 95% CIs. The size of each square reflects the weight of that comparison in the meta-analysis. The diamond represents the overall effect for all studies or within subgroups. The central vertical black line represents an RR of 1, and the dotted red line indicates the pooled RR from the meta-analysis. Overall, the RR for diagnosing brucellosis when compared to RBPT did not differ significantly [RR (95% CI) = 0.97 (0.81–1.17), *I*^2^ = 86%]. In subgroup analysis, slow agglutination tests had a significantly lower RR compared to RBPT [RR (95% CI) = 0.41 (0.25–0.68), *I*^2^ = 96%]. The CFT also demonstrated a slightly but significantly lower RR [RR (95% CI) = 0.97 (0.94–0.99), *I*^2^ = 9%]. However, the primary binding assays, PCR, and rapid agglutination tests did not differ significantly from RBPT [58,62,69,70,72,87,88,89,90,91,92,93,94,95,96,97,98,99,100,101,102,103,104,105,106,107,108,109,110,111,112,113,114,115,116].

**Figure 6 vetsci-12-00638-f006:**
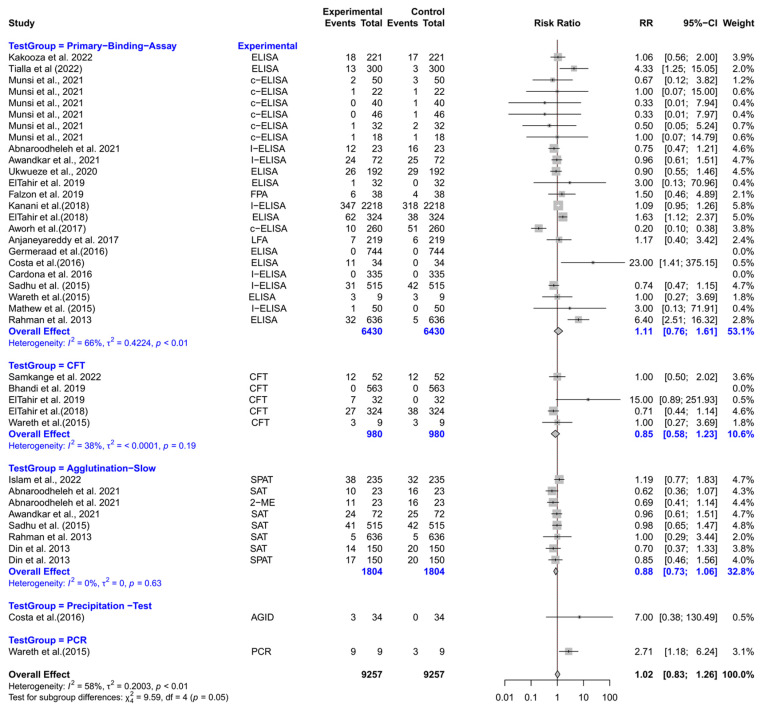
The forest plot represents the relative risk (RR) of 5 different test groups [experimental] compared with the Rose Bengal plate test (RBPT) [control] for diagnosing brucellosis in goats. A total of 39 comparisons are depicted in the forest plot. Three comparisons were excluded because they had zero positive events in both groups, i.e., experiment and control. In the forest plot, squares represent the point estimate (RR), and their associated horizontal lines indicate the respective 95% CIs. The size of each square reflects the weight of that comparison in the meta-analysis. The diamond represents the overall effect for all studies or within subgroups. The central vertical black line represents an RR of 1, and the dotted red line indicates the pooled RR from the meta-analysis. Overall, the RR did not differ significantly compared to RBPT [RR (95% CI) = 1.02 (0.83–1.26), *I*^2^ = 58%]. The subgroup analyses revealed that the primary binding assays, CFT, slow agglutination tests, precipitation tests, and PCR, did not differ significantly from RBPT [49,54,59,96,99,100,105,106,110,111,116,117,118,119,120,121,122,123,124,125,126,127,128].

**Figure 7 vetsci-12-00638-f007:**
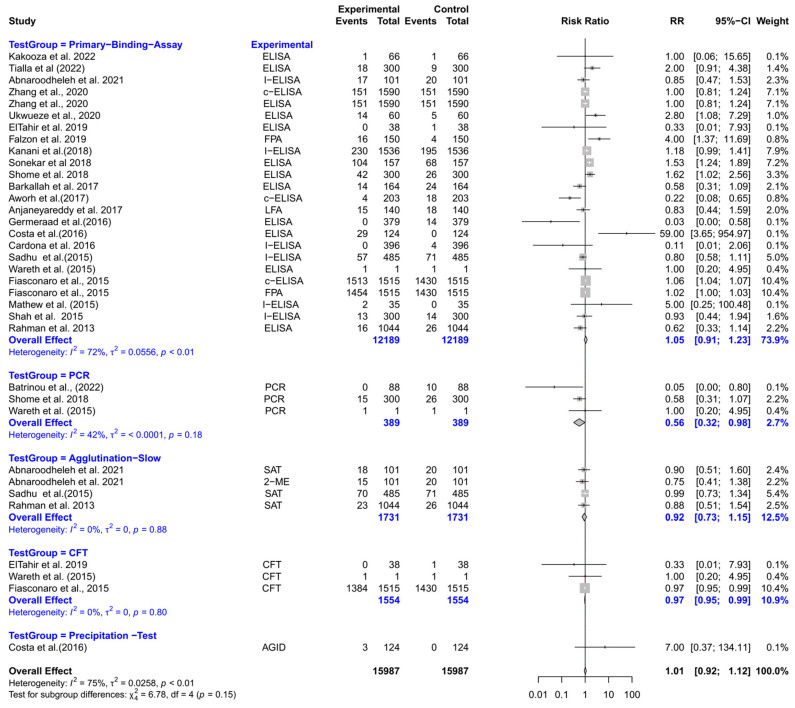
The forest plot represents the relative risk (RR) of 5 different test groups [experimental] compared with the Rose Bengal plate test (RBPT) [control] for diagnosing brucellosis in sheep. A total of 35 comparisons are depicted in the forest plot. In the forest plot, squares represent the point estimate (RR), and their associated horizontal lines indicate the respective 95% CIs. The size of each square reflects the weight of that comparison in the meta-analysis. The diamond represents the overall effect for all studies or within subgroups. The central vertical black line represents an RR of 1, and the dotted red line indicates the pooled RR from the meta-analysis. Overall, the RR did not differ significantly compared to RBPT [RR (95% CI) = 0.97 (0.86–1.09), *I*^2^ = 76%]. In subgroup analyses, PCR had a significantly lower RR compared to RBPT [RR (95% CI) = 0.56 (0.32–0.98), *I*^2^ = 42%]. The CFT also demonstrated a slightly but significantly lower RR [RR (95% CI) = 0.97 (0.95–0.99), *I*^2^ = 0%]. However, primary binding assays, slow agglutination tests, and precipitation tests did not differ significantly from RBPT [54,94,96,99,100,104,105,106,110,111,114,117,118,120,122,124,125,126,127,129,130,131].

**Figure 8 vetsci-12-00638-f008:**
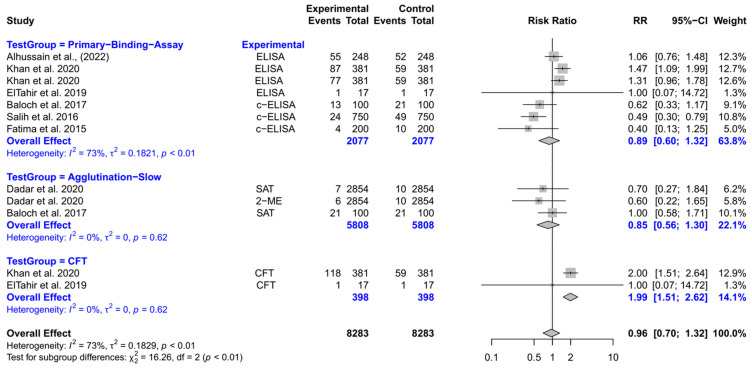
The forest plot represents the relative risk (RR) of 3 different test groups [experimental] compared with the Rose Bengal plate test (RBPT) [control] for diagnosing brucellosis in camels. A total of 12 comparisons are depicted in the forest plot. In the forest plot, squares represent the point estimate (RR), and their associated horizontal lines indicate the respective 95% CIs. The size of each square reflects the weight of that comparison in the meta-analysis. The diamond represents the overall effect for all studies or within subgroups. The central vertical black line represents an RR of 1, and the dotted red line indicates the pooled RR from the meta-analysis. Overall, the RR did not differ significantly compared to RBPT [RR (95% CI) = 0.96 (0.70–1.32), *I*^2^ = 73%]. The subgroup analyses revealed that the CFT had a significantly higher RR compared to RBPT [RR (95% CI) = 1.99 (1.51–2.62), *I*^2^ = 0%]. However, the primary binding assays and slow agglutination tests did not differ significantly from RBPT [3,99,132,133,134,135,136].

**Figure 9 vetsci-12-00638-f009:**
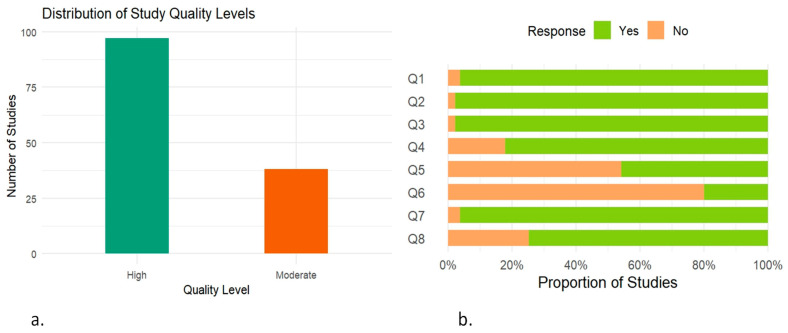
Assessment of study quality based on JBI critical appraisal checklist. (**a**) Bar chart showing the overall distribution of included studies categorized by their quality level. (**b**) Stacked bar chart illustrating the proportion of “Yes” and “No” responses for each of the eight questions (Q1–Q8); 1. Were the criteria for inclusion in the sample clearly defined? 2. Were the study subjects and the setting described in detail? 3. Was the exposure measured in a valid and reliable way? 4. Were objective, standard criteria used for measurement of the condition? 5. Were confounding factors identified? 6. Were strategies to deal with confounding factors stated? 7. Were the outcomes measured in a valid and reliable way? 8. Was appropriate statistical analysis used?

**Table 1 vetsci-12-00638-t001:** Summary of relative risk (RR) estimates of diagnostic test groups compared to RBPT in various species.

Species	Test Group Compared to RBPT	RR (95% CI)	*I*^2^ (%)	Interpretation
Human	Rapid agglutination tests	3.43 (1.78–6.59)	18	Higher risk
Human	Primary binding assays	1.75 (1.35–2.26)	73	Higher risk
Human	Slow agglutination tests	0.68 (0.48–0.96)	90	Lower risk
Human	PCR	1.45 (0.27–7.80)	95	Similar risk
Human	Culture	0.21 (0.02–2.85)	96	Similar risk
Cattle	Slow agglutination tests	0.41 (0.25–0.68)	96	Lower risk
Cattle	CFT	0.97 (0.94–0.99)	9	Slightly lower risk
Cattle	Primary binding assays	1.09 (0.91–1.30)	79	Similar risk
Cattle	PCR	1.34 (0.65–2.77)	92	Similar risk
Buffalo	Primary binding assays	1.11 (0.91–1.34)	0	Similar risk
Buffalo	PCR	11.16 (0.77–160.85)	68	Similar risk
Goat	Primary binding assays	1.11 (0.76–1.61)	66	Similar risk
Goat	CFT	0.85 (0.58–1.23)	38	Similar risk
Goat	Slow agglutination tests	0.88(0.73–1.06)	0	Similar risk
Sheep	Primary binding assays	1.05 (0.91–1.23)	72	Similar risk
Sheep	PCR	0.56 (0.32–0.98)	42	Lower risk
Sheep	Slow agglutination tests	0.92 (0.73–1.15)	0	Similar risk
Sheep	CFT	0.97 (0.95–0.99)	0	Slightly lower risk
Camel	Primary binding assays	0.89 (0.60–1.32)	73	Similar risk
Camel	Slow agglutination tests	0.85 (0.56–1.30)	0	Similar risk
Camel	CFT	1.99 (1.51–2.62)	0	Higher risk
Multi-species ruminants	CFT	0.95 (0.78–1.15)	15	Similar risk
Multi-species ruminants	Primary binding assays	1.04 (0.67–1.59)	95	Similar risk
Pig	Primary binding assays	1.46 (0.48–4.48)	94	Similar risk
Dog	Overall comparison with RBPT	1.70 (1.22–2.35)	48	Higher risk

## Data Availability

The data presented in this study are available on request from the corresponding author.

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
