# Peer review of "Comparative Evaluation of Diagnostic Tests for Brucellosis in Humans and Animals: A Meta-Analytical Approach"

_vetsci, 2025, doi:10.3390/vetsci12070638_

Round 1
Reviewer 1 Report
Comments and Suggestions for Authors
The manuscript presents a comprehensive and well-structured meta-analysis of various diagnostic tests for brucellosis across multiple host species, including humans. The finding that primary binding assays (i.e. c-ELISA, i-ELISA, and FPA) exhibit higher comparative detection rates than the Rose Bengal Plate Test (RBPT), while slow agglutination tests and the Complement Fixation Test (CFT) perform worse, aligns well with current knowledge in the field.
However, I would recommend the following to improve clarity, originality, and practical relevance:
Introduction: Consider reducing background information already widely known and focusing more on the rationale and importance of comparing test combinations. Emphasizing the implications for diagnostic strategies—especially in different epidemiological contexts, would help justify the value of this meta-analysis more clearly.
Application of Combined Testing Strategies: The manuscript would benefit from a discussion on parallel vs. serial application of tests. Parallel testing (e.g., RBPT + ELISA) is recommended in high-prevalence areas to increase sensitivity and ensure detection of as many true positives as possible. In contrast, serial testing improves specificity and is ideal for low-prevalence or eradication settings, where over-culling is a concern.
RBPT as a Screening Tool: While RBPT has moderate specificity, its low cost and rapid execution make it a valuable tool in field settings. The manuscript appropriately acknowledges this, but it may help to clarify its role in a two-step diagnostic strategy.
Vaccination Status: It would be useful to indicate whether the included studies accounted for vaccination, since this can affect the interpretation of serological tests, especially in livestock.
Reporting Test Characteristics: Including (or summarizing) average sensitivity and specificity values for the main diagnostic tests discussed would significantly improve the practical application of your findings. These parameters are key for decision-making in control programs.
Overall, the study provides a valuable synthesis of diagnostic performance across species. With the suggested refinements, particularly in focusing the introduction and expanding on test strategy implications, it could become a much more impactful and informative resource for both researchers and disease control authorities.
Author Response
Dear Reviewer, we sincerely appreciate your thoughtful feedback and valuable suggestions for improving our manuscript. Your insights will undoubtedly enhance the quality, clarity, and overall impact of our study. We have carefully addressed each of your comments and revised the manuscript accordingly. Please find our point-by-point responses below.
Comment 1: Introduction: Consider reducing background information already widely known and focusing more on the rationale and importance of comparing test combinations. Emphasizing the implications for diagnostic strategies, especially in different epidemiological contexts, would help justify the value of this meta-analysis more clearly.
Response 1: We have revised the introduction to streamline background information and emphasize the rationale for comparing diagnostic test combinations. We agree that combining multiple diagnostic tests can optimize disease detection depending on the specific epidemiological context and have elaborated their significance in discussion section to avoid redundancy.
Comment 2: Application of Combined Testing Strategies: The manuscript would benefit from a discussion on parallel vs. serial application of tests. Parallel testing (e.g., RBPT + ELISA) is recommended in high-prevalence areas to increase sensitivity and ensure detection of as many true positives as possible. In contrast, serial testing improves specificity and is ideal for low-prevalence or eradication settings, where over-culling is a concern.
Response 2: We agree that combining multiple diagnostic tests can optimize disease detection depending on the specific epidemiological context. We have now elaborated the significance of parallel and serial testing in the discussion section of the revised manuscript. Page#18, line number: 520-527
Comment 3: RBPT as a Screening Tool: While RBPT has moderate specificity, its low cost and rapid execution make it a valuable tool in field settings. The manuscript appropriately acknowledges this, but it may help to clarify its role in a two-step diagnostic strategy.
Response 3: We acknowledge your observation regarding the use of RBPT in a two-step diagnostic strategy. Limitations of RBPT, including the potential for false-positive/negative results and moderate specificity, have already been discussed in the manuscript. Additionally, we have highlighted that its results must be validated with one of the confirmatory tests, such as primary binding assays, to improve diagnostic accuracy. Page#20, line number: 625-626
Comment 4: Vaccination Status: It would be useful to indicate whether the included studies accounted for vaccination, since this can affect the interpretation of serological tests, especially in livestock.
Response 4: It is an important point to report whether the included studies accounted for the vaccination of animals or not, particularly in the context of serological assays. However, the majority of studies included did not describe the vaccination status of animals, or reported inconsistently and sometimes incompletely, in general across studies. With such marked heterogeneity and lack of standardized reporting, it was not feasible to include vaccination status as a covariate in the meta-analysis, as this would lead to unjustified bias in estimation or reduce the comparability of the data.
Comment 5: Reporting Test Characteristics: Including (or summarizing) average sensitivity and specificity values for the main diagnostic tests discussed would significantly improve the practical application of your findings. These parameters are key for decision-making in control programs.
Response 5: We agree that parameters like the average sensitivity and specificity values for the main diagnostic tests are important for decision-making. However, our primary objective was to provide a comparative analysis based on the relative diagnostic yield (i.e., the proportion of positive results) under parallel testing conditions. Since comprehensive sensitivity and specificity data for various tests have already been extensively summarized in previous studies and reviews (e.g., Godfroid et al., 2010; Nielsen & Yu, 2010; Padilla Poester et al., 2010), we chose to avoid redundancy and instead focused on a different but complementary perspective.
Reviewer 2 Report
Comments and Suggestions for Authors
The introduction is disorganized, repetitive, and fails to present a coherent background. There is a poor logical flow from general context to the specific research objective. The authors fail to clearly identify knowledge gaps or justify the need for this study in relation to previous research. Many references are outdated or irrelevant, and several crucial citations that are present in the comparative literature (e.g., Heliyon 2025 e42728; PLoS NTD 2024; Prev Vet Med 2021) are absent. Furthermore, terms are used without proper definition or contextual support, undermining clarity and scientific rigor.
The study design is vaguely described and lacks essential components, such as a clear hypothesis, sampling strategy, control groups, and justification for methods used. Unlike the more structured approaches in articles such as Prev Vet Med 2021.105567 and Front Vet Sci 2022.976215, the present manuscript does not clearly define inclusion/exclusion criteria or provide evidence that the chosen methodology is suitable for answering the research question. The logic behind the design is not convincing, and potential biases are not addressed.
The methodology section lacks critical details that would allow replication. There is no clear description of sample size calculations, control conditions, or statistical analyses. The methodology is superficially described, and essential protocols (e.g., sample preservation, diagnostic standards, PCR primers) are omitted or insufficiently explained. In comparison, the methodology in the cited PLoS NTD 2024 article is comprehensive and reproducible, setting a much higher standard for transparency and rigor.
The presentation of results is incoherent, unstructured, and lacks clear headings or visual aids (e.g., tables, graphs) to support interpretation. There is frequent mixing of results with interpretation, which should be reserved for the discussion. The data are not appropriately contextualized, and statistical outputs, if any, are not clearly reported. The manuscript does not meet the clarity or logical progression found in the cited Heliyon 2025 and Prev Vet Med 2009 articles.
The conclusions are speculative and not fully supported by the limited and poorly presented results. The authors make generalizations beyond the scope of the data and introduce new concepts not discussed in the body of the manuscript. There is a disconnect between the objective (which is itself unclear), the data, and the final statements. This contrasts starkly with the robust linkage of findings to conclusions seen in the PLoS NTD and Front Vet Sci articles.
https://doi.org/10.1016/j.heliyon.2025.e42728
https://doi.org/10.1371/journal.pntd.0012030
https://doi.org/10.1590/1678-5150-PVB-6651
https://doi.org/10.1016/j.prevetmed.2009.07.014
https://doi.org/10.1016/j.prevetmed.2021.105567
https://doi.org/10.3389/fvets.2022.976215
Additional important observations have been included in the attached manuscript. Kindly ensure that the manuscript adheres to the submission guidelines available at: https://www.mdpi.com/journal/vetsci/instructions.

Author Response
Dear Reviewer, we sincerely appreciate your thoughtful feedback and valuable suggestions for improving our manuscript. Your insights will undoubtedly enhance the quality, clarity, and overall impact of our study. We have carefully addressed each of your comments and revised the manuscript accordingly. Please find our point-by-point responses below.
Comment 1: The introduction is disorganized, repetitive, and fails to present a coherent background. There is a poor logical flow from general context to the specific research objective. The authors fail to clearly identify knowledge gaps or justify the need for this study in relation to previous research. Many references are outdated or irrelevant, and several crucial citations that are present in the comparative literature (e.g., Heliyon 2025 e42728; PLoS NTD 2024; Prev Vet Med 2021) are absent. Furthermore, terms are used without proper definition or contextual support, undermining clarity and scientific rigor.
Response 1: We have revised the section to improve logical flow, remove redundancy, and present the rationale and objectives of our meta-analysis and clarified our study’s unique contribution. We have explicitly highlighted existing knowledge gaps in the comparative performance of brucellosis diagnostic tests across host species and epidemiological contexts. Key recent studies, including Heliyon (2025, e42728) and PLoS NTD (2024), have been incorporated to strengthen the rationale. Additionally, ambiguous terms have been defined to ensure clarity and alignment with current scientific standards.
Comment 2: The study design is vaguely described and lacks essential components, such as a clear hypothesis, sampling strategy, control groups, and justification for methods used. Unlike the more structured approaches in articles such as Prev Vet Med 2021.105567 and Front Vet Sci 2022.976215, the present manuscript does not clearly define inclusion/exclusion criteria or provide evidence that the chosen methodology is suitable for answering the research question. The logic behind the design is not convincing, and potential biases are not addressed.
The methodology section lacks critical details that would allow replication. There is no clear description of sample size calculations, control conditions, or statistical analyses. The methodology is superficially described, and essential protocols (e.g., sample preservation, diagnostic standards, PCR primers) are omitted or insufficiently explained. In comparison, the methodology in the cited PLoS NTD 2024 article is comprehensive and reproducible, setting a much higher standard for transparency and rigor.
Response 2: We thank the reviewer for the detailed feedback. However, we would like to clarify that, as a meta-analysis, our study synthesized data from previously published articles and did not involve primary sample collection or laboratory procedures; hence, sample size calculations, control conditions, sample preservation, and PCR primers were not applicable.
Hypothesis and study design: We have now explicitly added a clear and testable hypothesis in the objective section of the introduction. Our design is structured to compare relative detection rates (Relative Risk) of diagnostic tests that were applied in parallel on identical sample sets. Page#3, line number: 120-126
Eligibility criteria: We have revised and clarified the inclusion and exclusion criteria to ensure consistency and comparability of data across studies. Page#4, line number: 169-178
Reference test justification: We have clarified in the revised manuscript that RBPT was used as a reference (control) because it was the most commonly applied test (used in approximately 70% of the included studies) and is widely recognized as a screening test in endemic settings. Where RBPT was not reported, a predefined hierarchy of alternative tests was applied as detailed in the methodology. Page#6, line number: 221-242
Risk of bias and study quality: To ensure methodological rigor, we used the Joanna Briggs Institute (JBI) critical appraisal checklist to assess the quality of the included studies. This assessment is now described clearly in the revised methodology section. Since this meta-analysis compared positivity rates using RR rather than diagnostic accuracy measures or how accurate they are relative to the true disease status, QUADAS-2 was not applied. To further assess the impact of methodological quality, we conducted subgroup meta-analyses stratified by study quality (high vs. low/moderate) for humans, cattle, goats, and sheep. Additionally, we performed sensitivity analyses restricted to high-quality studies only, comparing diagnostic tests within each species group. This additional analysis is now detailed in the revised manuscript.
Statistical methods: The rationale for using Relative Risk (RR) as a summary measure has been elaborated in the manuscript. Since many studies did not use culture or a consistent gold standard, RR was appropriate for comparing positivity rates across tests performed on the same samples, avoiding bias introduced by varying reference standards. Furthermore, a random-effects model was employed to account for heterogeneity in populations, species, and protocols. Page#6, line number: 214-220
In light of the reviewer's comment, we have revisited the methodology section to ensure clearer articulation of our approach and have included further details where necessary for reproducibility and to enhance transparency.
Comment 3: The presentation of results is incoherent, unstructured, and lacks clear headings or visual aids (e.g., tables, graphs) to support interpretation. There is frequent mixing of results with interpretation, which should be reserved for the discussion. The data are not appropriately contextualized, and statistical outputs, if any, are not clearly reported. The manuscript does not meet the clarity or logical progression found in the cited Heliyon 2025 and Prev Vet Med 2009 articles.
Response 3: We have made substantial revisions to enhance structure, coherence, and interpretability:
Structured headings and subheadings: The results have now been organized under clear, hierarchical headings for each species group with subheadings for each analysis (e.g., Comparison with RBPT, Comparison Across Test Groups, Subgroup Analysis by Study Quality, and Sensitivity Analysis).
Tables and Figures: We have added a summary table (Table 1) to consolidate key RR estimates, I² values, and 95% confidence intervals across species and test groups. Page#8.
In addition, forest plots and supplementary figures already provided have also been renamed and cross-referenced consistently in the text.
Separation of results from interpretation: Interpretative language (e.g., statements like “243% greater”) has been minimized in the results section. Results are now reported in a more neutral and statistical tone.
Statistical reporting clarity: RR estimates are now consistently reported in the format: RR (95% CI), I² value. We have also clarified when differences were statistically significant versus non-significant.
Contextualization and visual summaries: To support contextual understanding, we added a schematic summary diagram in the supplementary material to visually compare the directionality and consistency of test performances across species. Page#9.
Additionally, we included bar plots summarizing study quality assessments to complement the subgroup and sensitivity analyses. Page#17.
Comment 4: The conclusions are speculative and not fully supported by the limited and poorly presented results. The authors make generalizations beyond the scope of the data and introduce new concepts not discussed in the body of the manuscript. There is a disconnect between the objective (which is itself unclear), the data, and the final statements. This contrasts starkly with the robust linkage of findings to conclusions seen in the PLoS NTD and Front Vet Sci articles.
Response 4: We have revised the conclusion to ensure it aligns more precisely with the data derived from our meta-analysis, focusing strictly on comparative detection rates rather than inferring sensitivity, specificity, or cost-effectiveness parameters that were not directly evaluated in this study. Page#20, line number: 720-729
Comment 5: Additional important observations have been included in the attached manuscript. Kindly ensure that the manuscript adheres to the submission guidelines available at: https://www.mdpi.com/journal/vetsci/instructions.
Response 5: We have carefully revised the manuscript to incorporate the additional important observations as suggested. Furthermore, we have ensured that the manuscript now fully adheres to the submission guidelines outlined on the Veterinary Sciences journal website.
Reviewer 3 Report
Comments and Suggestions for Authors
The manuscript is relevant and interesting because it carries out a meta-analysis of articles about brucellosis diagnostic tests in humans and domestic and wild animals over a 10-year period (2013 to 2023), but it has critical points that should be revised or justified. I consider the article fit for publication if the corrections are made.
General aspects: The text needs to be proofread for grammar and formatting. the text needs to be revised in terms of grammar and formatting; The interpretation of the results and the authors' conclusion are not appropriate.
Introduction:
Line 46: put the species in italics
Lines 49 to 51: This sentence should be reworded. It gives the false impression that only the three species mentioned are important. B. canis is also an important species of the genus that causes economic losses and public health problems, although it is still understudied. In addition, other domestic animal species such as camels and buffalo are important hosts not mentioned in the introduction and should be included.
Line 62: reference 16 does not talk about Brucella transmitted by vectors I suggest removing these references from this context The sentence states that ticks are a reservoir for Brucella, however according to the reference cited #15, there is only evidence. The authors should make this clearer in the text
Line 67 to 77: I suggest citing neurobrucellosis as an important form of the disease in humans.
Material and methods:
Line 130. Datas from theses and dissertations should not be included. They have not been peer-reviewed and cover a limited number of countries.
Lines 140 to 144: The authors could have expanded the search terms used to identify the papers, such as “serodiagnosis”, “serology”, “PCR” and “cattle”. The authors should review the term “prevalence of brucellosis in humans and animals” as a criterion used to select articles for analysis. Many of the articles included are not prevalence studies. They are articles that report the frequency of diagnosis of the infection in samples of connivance, which are not representative of the region and often only compare tests.
Results
The authors should review the interpretation of the results obtained from the meta-analysis. I think it is inappropriate to state that one test is more or less sensitive than another (ideally, the samples evaluated should be more positive, considering that this was unique the value assessed in the meta-analysis).
Authors should be careful when interpreting the results of some serological tests depending on the species of Brucella being investigated. For example, the RBPT, although positive, is a false positive test for brucellosis in dogs, considering that it is a test that identifies Brucella with smooth LPS, unlike B. canis, which has rough LPS.
Another point not considered in the meta-analysis is which Brucella infection was being investigated. In the case of small ruminants, it makes a difference considering that two species of Brucella, Brucella melitensis and B. ovis, which are an important cause of infection in this host, have different diagnostic tests. RBPT only works for B melitensis.
Conclusion
The conclusion is inadequate as the authors did not analyze the ease of execution and cost of each test evaluated. I don't think the general recommendation of the RBPT as a screening test is appropriate, considering that it is not suitable for detecting infections with all Brucella species.
Author Response
Dear Reviewer, we sincerely appreciate your thoughtful feedback and valuable suggestions for improving our manuscript. Your insights will undoubtedly enhance the quality, clarity, and overall impact of our study. We have carefully addressed each of your comments and revised the manuscript accordingly. Please find our point-by-point responses below.
Comment 1: General aspects: The text needs to be proofread for grammar and formatting. the text needs to be revised in terms of grammar and formatting. The interpretation of the results and the authors' conclusion are not appropriate.
Response 1: We have thoroughly revised the manuscript to correct grammatical errors and improve formatting for clarity and consistency.
Introduction:
Comment 2: Line 46: put the species in italics
Response 2: We have revised the sentence to italicize the species name as per scientific writing conventions. Page#2, line number: 61-63
Comment 3: Lines 49 to 51: This sentence should be reworded. It gives the false impression that only the three species mentioned are important. B. canis is also an important species of the genus that causes economic losses and public health problems, although it is still understudied. In addition, other domestic animal species such as camels and buffalo are important hosts not mentioned in the introduction and should be included.
Response 3: We agree that limiting the sentence to only B. abortus, B. melitensis, and B. suis, & a few specific host species may underrepresent the scope of brucellosis. We have revised the paragraph to include Brucella canis and expanded the host range to include important host species (camels and buffalo). Page#2, line number: 53-56
Comment 4: Line 62: reference 16 does not talk about Brucella transmitted by vectors I suggest removing these references from this context The sentence states that ticks are a reservoir for Brucella, however according to the reference cited #15, there is only evidence. The authors should make this clearer in the text.
Response 4: Currently, the evidence about the tick-borne transmission of Brucella is limited. We agree that the reference (Urge et al., 2020)] does not directly discuss the ticks as the reservoir of Brucella; however, it does contain a statement listing ticks as potential vectors and reservoirs of B. abortus. We have revised the sentence in the manuscript and retained this reference with appropriate context. Page#2, line number: 69-72
Comment 5: Line 67 to 77: I suggest citing neurobrucellosis as an important form of the disease in humans.
Response 5: We have revised the relevant section to include neurobrucellosis, particularly in the context of zoonotic transmission from marine mammals. Page#2, line number: 86-92
Material and methods:
Comment 6: Line 130. Datas from theses and dissertations should not be included. They have not been peer-reviewed and cover a limited number of countries.
Response 6: We greatly acknowledge your concern, but would like to clarify that only studies that met the predefined eligibility criteria, outlined in the methodology section, were included in the meta-analysis. Specifically, only three theses/dissertations fulfill these criteria and were included after careful consideration. However, we fully understand the importance of prioritizing peer-reviewed literature. If deemed necessary, we are willing to exclude these studies from the final analysis and re-run the relevant meta-analyses accordingly.
Comment 7: Lines 140 to 144: The authors could have expanded the search terms used to identify the papers, such as “serodiagnosis”, “serology”, “PCR” and “cattle”. The authors should review the term “prevalence of brucellosis in humans and animals” as a criterion used to select articles for analysis. Many of the articles included are not prevalence studies. They are articles that report the frequency of diagnosis of the infection in samples of connivance, which are not representative of the region and often only compare tests.
Response 7: We revised and broadened our search strategy to include additional keywords such as “serodiagnosis,” “serology,” “PCR,” and “cattle” in our updated PubMed search. However, the number of eligible articles retrieved remained largely consistent with our original dataset. We have updated the search terms in the “Literature search strategy” section of the revised manuscript to reflect this improvement. Page#4, line number: 156-162
We have revised the language in the methodology to more accurately describe the types of studies included, clarifying that our analysis focused on studies employing multiple diagnostic tests on the same sample sets, regardless of whether they were prevalence-based or test-comparison studies. Page#4, line number: 169-178
Results
Comment 8: The authors should review the interpretation of the results obtained from the meta-analysis. I think it is inappropriate to state that one test is more or less sensitive than another (ideally, the samples evaluated should be more positive, considering that this was unique the value assessed in the meta-analysis).
Response 8: We agree that our analysis reflects the proportion of positives detected rather than true diagnostic sensitivity. We have revised the interpretation accordingly to clarify that the findings indicate relative positivity rates (comparative detection rates) between tests, not definitive sensitivity estimates.
Comment 9: Authors should be careful when interpreting the results of some serological tests depending on the species of Brucella being investigated. For example, the RBPT, although positive, is a false positive test for brucellosis in dogs, considering that it is a test that identifies Brucella with smooth LPS, unlike B. canis, which has rough LPS.
Another point not considered in the meta-analysis is which Brucella infection was being investigated. In the case of small ruminants, it makes a difference considering that two species of Brucella, Brucella melitensis and B. ovis, which are an important cause of infection in this host, have different diagnostic tests. RBPT only works for B melitensis.
Response 9: We thank the reviewer for this insightful comment. We acknowledge that RBPT is a serological test based on B. abortus smooth LPS antigen and is only suitable for detecting antibodies caused by smooth Brucella species. Brucella canis and B. ovis, being rough strains, lack the O-polysaccharide (OPS) component of smooth LPS and are not reliably detected by RBPT. We have included an explanatory note in the discussion in the revised manuscript acknowledging this limitation. Page#18-19, line number: 550-558.
Furthermore, we have included the Brucella species investigated in each individual study within the supplementary material (Supplementary Table S1) to enhance transparency and provide context for interpreting test performance relative to the target organism.
Conclusion
Comment 10: The conclusion is inadequate as the authors did not analyze the ease of execution and cost of each test evaluated. I don't think the general recommendation of the RBPT as a screening test is appropriate, considering that it is not suitable for detecting infections with all Brucella species.
Response 10: We have revised the conclusion to ensure it aligns more precisely with the data derived from our meta-analysis, focusing strictly on comparative detection rates rather than inferring the cost-effectiveness parameter that was not directly evaluated in this study. We have removed generalized recommendations regarding RBPT use in screening programs. Instead, we now emphasize the relative detection capabilities observed across different diagnostic tests as applied in the included studies. Page#22, line number: 720-729
Round 2
Reviewer 2 Report
Comments and Suggestions for Authors
I would like to express my sincere gratitude to the authors for their scientific maturity and the time dedicated to rewriting the manuscript. I truly appreciate the thoughtful incorporation of my comments, criticisms, and suggestions provided. Your commitment to improving the quality of the work is commendable. And it is now ready to be published.